# Health system assessment for access to care after injury in low- or middle-income countries: A mixed methods study from Northern Malawi

John Whitaker[1,2,3]*, Idara Edem[2,4,5], Ella Togun[2], Abena S. Amoah[6,7], Albert Dube[6], Lindani Chirwa[8,9], Boston Munthali[10,11], Giulia Brunelli[12], Thomas Van Boeckel[12,13], Rory Rickard[3], Andrew JM Leather[2‡], Justine Davies[1,14,15‡]

**1** Institute of Applied Health Research, University of Birmingham, Birmingham, United Kingdom, **2** King's Centre for Global Health and Health Partnerships, School of Life Course and Population Sciences, Faculty of Life Sciences and Medicine, King's College London, London, United Kingdom, **3** Academic Department of Military Surgery and Trauma, Royal Centre for Defence Medicine, Birmingham, United Kingdom, **4** Insight Institute of Neurosurgery & Neuroscience, Flint, Michigan, United States of America, **5** Michigan State University, East Lansing, Michigan, United States of America, **6** Malawi Epidemiology and Intervention Research Unit (formerly Karonga Prevention Study), Chilumba, Malawi, **7** Faculty of Epidemiology and Population Health, London School of Hygiene & Tropical Medicine, London, United Kingdom, **8** Karonga District Hospital, Karonga District Health Office, Karonga, Malawi, **9** School of Medicine & Oral Health, Department of Pathology, Kamuzu University of Health Sciences (KUHeS), Blantyre, Malawi, **10** Mzuzu Central Hospital, Department of Orthopaedic Surgery, Mzuzu, Malawi, **11** Lilongwe Institute of Orthopaedic and Neurosurgery, Lilongwe, Malawi, **12** Health Geography and Policy Group, ETH Zurich, Zurich, Switzerland, **13** Center for Disease Dynamics Economics and Policy, Washington, DC, United States of America, **14** Medical Research Council/Wits University Rural Public Health and Health Transitions Research Unit, Faculty of Health Sciences, School of Public Health, University of the Witwatersrand, Johannesburg, South Africa, **15** Department of Global Surgery, Stellenbosch University, Stellenbosch, South Africa

‡ These authors are joint senior authors on this work.
* j.whitaker@bham.ac.uk

**Data Availability Statement:** All data relevant to this report's analysis are within the manuscript and supporting information files.

## Abstract

### Background

Injuries represent a vast and relatively neglected burden of disease affecting low- and middle-income countries (LMICs). While many health systems underperform in treating injured patients, most assessments have not considered the whole system. We integrated findings from 9 methods using a 3 delays approach (delays in seeking, reaching, or receiving care) to prioritise important trauma care health system barriers in Karonga, Northern Malawi, and exemplify a holistic health system assessment approach applicable in comparable settings.

### Methods and findings

To provide multiple perspectives on each conceptual delay and include data from community-based and facility-based sources, we used 9 methods to examine the injury care health system. The methods were (1) household survey; (2) verbal autopsy analysis; (3) community focus group discussions (FGDs); (4) community photovoice; (5) facility care-pathway process mapping and elucidation of barriers following injury; (6) facility healthcare worker

**Funding:** JW was awarded funding from the Drummond Committee of the Royal Army Medical Corps Charity (https://www.ramcassociation.org.uk/about/the-role-of-the-rhq-ramc), the Royal College of Surgeons of England research fellowship scheme (https://www.rcseng.ac.uk/standards-and-research/research/fellowships-awards-grants/fellowships/one-year-surgical-research-fellowship/) and the King's centre for global health partnerships (https://www.kcl.ac.uk/kghp). There are no specific grant numbers associated with these awarded funds. The funders had no role in study design, data collection and analysis, decision to publish, or preparation of the manuscript.

**Competing interests:** The authors have declared that no competing interests exist.

**Abbreviations:** CHAM, Christian Health Association of Malawi; FGD, focus group discussion; GIS, geographic information system; GRAMMS, Good Reporting of A Mixed Methods Study; HDSS, Health and Demographic Surveillance Site; LCoGS, Lancet Commission on Global Surgery; LMICs, low- and middle-income countries; MEIRU, Malawi Epidemiology and Intervention Research Unit.

survey; (7) facility assessment survey; (8) clinical vignettes for care process quality assessment of facility-based healthcare workers; and (9) geographic information system (GIS) analysis. Empirical data collection took place in Karonga, Northern Malawi, between July 2019 and February 2020. We used a convergent parallel study design concurrently conducting all data collection before subsequently integrating results for interpretation. For each delay, a matrix was created to juxtapose method-specific data relevant to each barrier identified as driving delays to injury care. Using a consensus approach, we graded the evidence from each method as to whether an identified barrier was important within the health system.

We identified 26 barriers to access timely quality injury care evidenced by at least 3 of the 9 study methods. There were 10 barriers at delay 1, 6 at delay 2, and 10 at delay 3. We found that the barriers "cost," "transport," and "physical resources" had the most methods providing strong evidence they were important health system barriers within delays 1 (seeking care), 2 (reaching care), and 3 (receiving care), respectively. Facility process mapping provided evidence for the greatest number of barriers—25 of 26 within the integrated analysis. There were some barriers with notable divergent findings between the community- and facility-based methods, as well as among different community- and facility-based methods, which are discussed. The main limitation of our study is that the framework for grading evidence strength for important health system barriers across the 9 studies was done by author-derived consensus; other researchers might have created a different framework.

## Conclusions

By integrating 9 different methods, including qualitative, quantitative, community-, patient-, and healthcare worker-derived data sources, we gained a rich insight into the functioning of this health system's ability to provide injury care. This approach allowed more holistic appraisal of this health system's issues by establishing convergence of evidence across the diverse methods used that the barriers of cost, transport, and physical resources were the most important health system barriers driving delays to seeking, reaching, and receiving injury care, respectively. This offers direction and confidence, over and above that derived from single methodology studies, for prioritising barriers to address through health service development and policy.

## Author summary

### Why was this study done?

- Injuries represent a vast and relatively neglected burden of disease affecting low- and middle-income countries (LMICs).

- While evidence suggests that many health systems underperform in treating injured patients in LMICs, most assessments have not considered all elements of the healthcare system for injured people.

- Innovative mixed methods approaches to holistic health system assessment including community and facility perspectives are therefore needed.

## What did the researchers do and find?

- To examine the injury care health system in Karonga, Northern Malawi, we integrated the findings from 9 different methods: (1) household survey; (2) verbal autopsy analysis; (3) community focus group discussions (FGDs); (4) community photovoice; (5) facility care-pathway process mapping and elucidation of barriers following injury; (6) facility healthcare worker surveys; (7) facility assessment surveys; (8) clinical vignettes for care quality assessment of facility-based healthcare workers; and (9) geographic information system (GIS) analysis.

- We graded the strength of evidence each method provided as to whether a given barrier was important in inhibiting access to timely quality injury care.

- We found 26 barriers evidenced by at least 3 of the 9 methods with the barriers "cost," "transport," and "physical resources" having the strongest evidence that they were important barriers delaying seeking, reaching, and receiving care, respectively.

## What do these findings mean?

- By comparing the findings from the perspectives of 9 different methods, we were able to gain an in-depth understanding of the health system for trauma care.

- This approach can allows researchers and planners to know the barriers consistently shown to be important and prioritise health service development and policy interventions accordingly.

- This study, to our knowledge, represents a novel and innovative approach in terms of both the number and types of methods mixed, serving as an example other researchers could use in similar contexts.

- The way we graded evidence strength for comparison across methods was somewhat subjective and other researchers may have made different judgements.

## Introduction

Injuries resulting from trauma represent a vast and relatively neglected burden of disease, predominantly affecting low- and middle-income countries (LMICs) [1–3]. To ensure injured people can seek, reach, and receive high-quality care when needed requires a whole health system response [4]. However, while studies have shown that health systems underperform in treating injured patients in LMICs [1,5–7], a systematic review of the literature has shown most published assessments have not considered the whole system [8].

Health systems are complex adaptive systems representing a challenging or "wicked" problem to understand [9]. They are built out of historical contexts, rooted within human institutions, influenced by social behaviours governing function and performance, and unpredictably responsive to interventions [10–12]. Meaningful assessments should therefore include the whole health system if the desired outcome is to support impactful health system improvement [13–15].

Systems thinking has encouraged moves away from understanding health systems only through positivist or reductionist approaches [11,16,17]. Constructivist paradigms, which seek

to understand a health system's elements within a social and ecological context, are necessary. Well-designed studies that incorporate mixed method approaches [18], combining the strengths of different underpinning research philosophies, are needed [19]. Such research can allow a better-developed and more nuanced understanding of a problem than is possible through single methodological approaches alone [20].

Many different frameworks exist for describing, understanding, and researching health systems as complex systems with origins in differing paradigms of understanding and sociopolitical backgrounds [21]. A universal framework for understanding complex systems may therefore not exist, and tools selected for a particular assessment should match the intended purpose [21]. The 3 delays framework, which specifically considers factors delaying care seeking (delay 1), reaching a place of care (delay 2), and receiving appropriate and quality care (delay 3) [22], has been widely adopted across reproductive, maternal, newborn, and child health [23–27]. The Lancet Commission on Global Surgery (LCoGS) popularised the framework's use in broader surgical research on barriers in access to quality care [28], and, subsequently, it has become an accepted framework used to evaluate emergency healthcare in LMICs, including trauma [5,29–31].

We used a 3 delays framed mixed method assessment to achieve a well-developed understanding of the whole trauma care health system in Karonga, Northern Malawi. We aimed to integrate findings from 9 diverse methods to show important health system barriers to accessing quality care and thereby exemplify a holistic approach to health system assessment applicable in comparable low-income settings.

## Methods

### Study setting

Malawi is a landlocked, low-income country; the fourth poorest in the world [32]. Half of its population are children, and 70% of the population lives on less that $2.15 a day [32,33]. Although limited, available evidence confirms, like in many LMICs, injury is a serious and increasingly important health problem causing substantial and potentially avoidable mortality and morbidity [34]. Most studies on injury care within Malawi have focused on facility-based care [35–37]. There is little evidence available about prehospital care, transport, and reaching care and almost no understanding of population barriers to seeking care following injury, in keeping with the limited global literature on this subject [8].

The study was conducted at the Karonga Health and Demographic Surveillance Site (HDSS) of the Malawi Epidemiology and Intervention Research Unit (MEIRU). It focused specifically on the health system serving the population of the Karonga HDSS in Karonga District, Northern Malawi [38]. The HDSS has a population of over 40,000 (10,228 households) under surveillance since 2002 [38]. It is based in the surrounds of Chilumba, in the south of Karonga, a predominantly rural lakeshore district, typical of a Malawian subsistence economy community [38]. The HDSS population is mostly rural, although about 15% live in semi-urban settlements. Half of the population lives within 1 km of a paved road [38]. The main paved road runs through the district with mostly gravel secondary roads.

The Malawian government provide free facility healthcare to all residents, although out-of-pocket household payments still occur [39]. Traditional healers are recognised as deliverers of health services, although there is little link with the formal health system [40]. The private sector consists of private for-profit and private not-for-profit providers, mainly the Christian Health Association of Malawi (CHAM), which trains healthcare workers and provides services particularly for populations remote from public facilities [41]. The Karonga HDSS population is served by local primary facilities run by the government, including a military facility

accessible by civilians, private-for-profit, and CHAM providers. Secondary care facilities include a government facility 70 km north and a CHAM facility 40 km south over difficult hilly terrain. Tertiary care is provided at a government facility in the regional capital, Mzuzu, 150 km south. Most clinicians are non-physicians [42]. Medical Assistants, Clinical Officers, and doctors have completed post-secondary training of a 2-year clinical medicine certificate, 3-year clinical medicine diploma, and 6-year Bachelor's degree, respectively [35]. All provide immediate care to injured patients. Primary facilities are usually staffed by Medical Assistants and Clinical Officers. There is no established prehospital emergency medical service.

## Study design

The study design was created following a systematic literature review on trauma health system assessment methods [8], informal literature searches on health system assessment approaches used for other health conditions [43–49], and discussion among coauthors. A range of methods were chosen to holistically assess the health system in its ability to provide injury care. Hence, they cover multiple aspects of care provision and access and multiple perspectives, from community to facility facing. A prior Delphi study [30] to elucidate barriers at each of the 3 delay stages (delay 1—seeking care, delay 2—reaching a place of care, and delay 3—receiving appropriate and quality care) for injury also informed discussions between authors on which methodologies to use to assess those barriers and shaped the design of those methodologies to capture these barriers.

Nine methods to examine the injury care health system were agreed upon which captured all relevant information and were feasible to deliver within MEIRU. These were selected to provide multiple perspectives on each conceptual delay, encompass a range of methodological approaches, and include data from community-based and facility-based sources. The study methods were (1) household survey; (2) verbal autopsy analysis; (3) community focus group discussions (FGDs); (4) community photovoice; (5) facility care-pathway process mapping and elucidation of barriers following injury; (6) facility healthcare worker survey; (7) facility assessment survey; (8) clinical vignettes for care process quality assessment of facility-based healthcare workers; and (9) geographic information system (GIS) analysis. A more detailed description of unpublished study methods is reported in S1 Text.

Study methods involving discussion or interview with educated healthcare workers were conducted in English by JW. Study methods based in the community or involving survey or discussion with less educated healthcare workers (for example, security staff) were conducted in the vernacular language (Chitumbuka). A summary of the study methods included in the mixed methods assessment is in Table 1. Data collection took place between July 2019 and February 2020.

## Mixed methods analysis

We used a mixed methods approach and followed the Good Reporting of A Mixed Methods Study (GRAMMS) principles for reporting mixed methods research [50]. We adopted a pragmatic philosophical approach to mixing methods, using a convergent parallel study design concurrently conducting all data collection before subsequently integrating results for interpretation [51,52]. We adopted this approach to allow interpretation of the convergence, divergence, silence, or other relationships of the findings [52–55].

Many study methods were constructed and analysed with the 3 delays framework specifically in mind using a framework of barriers to injury care derived from a previously published Delphi study [30]. This Delphi study developed expert consensus on the most important barriers to consider when evaluating an LMIC health system's preparedness to deliver injury care

**Table 1. Summary of individual health system assessment study methods used.** The details of focus, purpose, sample size, and which delay they assessed are included. The table aims to summarise the methods with more detail available in either the associated publication referenced, or S1 Text.

| Method | Community or facility based | Purpose | Sample size achieved | Delay assessing |
|---|---|---|---|---|
| Household survey | Community | Nonfatal injury burden description and assessment of health system utilisation. | 1,819 households participated. | 1, 2, 3 |
| Verbal autopsy | Community | Assessment of avoidability of fatal injuries within the population and barriers associated with delays to care. | 185 injury deaths | 1, 2, 3 |
| FGD | Community | Community perspectives on the barriers driving delays to injury care. | 3 FGDs of 7 participants each | 1, 2, 3 |
| Community photovoice | Community | Community perspectives on the barriers driving delays to injury care. | 1 group of 7 participants | 1, 2, 3 |
| Facility process mapping [83] | Facility | Healthcare worker descriptions of the process of seeking, reaching and receiving care following injury, and any barriers driving delays. | 11 facility maps (54 healthcare workers in total). | 1, 2, 3 |
| Healthcare worker survey | Facility | Healthcare worker perspectives on the barriers driving delays to seeking reaching and receiving care following injury. | 228 healthcare workers surveyed | 1, 2, 3 |
| Facility assessment survey | Facility | Assessment of the resources available at each facility to manage injured patients. | 11 facility assessment surveys | 3 |
| Clinical vignettes [96] | Facility | Assessment of the quality of care process provided to injured patients. | 85 clinicians | 3 |
| GIS analysis [93] | Not applicable | Description of proximity and physical accessibility of health facilities to the population. | Not Applicable | 2 |

FGD, focus group discussion; GIS, geographic information system.

[30]. Additional barriers generated inductively from at least 3 or more individual study methods were also included. For each delay, a matrix was created to juxtapose the study-specific data relevant to each barrier (Table 2). To assess these data for convergence (the extent to which findings complement or reinforce each other [20]), a consensus exercise was undertaken between 3 authors (JW, IE, and AD). These authors discussed a framework for how barriers identified by each method would be graded according to the evidence provided that they were an important health system barrier. For this purpose, importance of the identified health system barrier was considered a function of both (a) numbers of people potentially affected and (b) the amount of harm potentially caused. Using this framework agreed by these 3 authors, each barrier was graded as having either strong, moderate, or weak evidence that it was important within the health system. Barriers for which a study methodology provided no evidence were categorised as silent. This silence could be a limitation of a method being unsuitable to detect evidence for a barrier, or suggestive that the barrier does not exist. This framework for grading the strength of evidence from each method that a barrier is important is detailed in Table 3. This approach to integrating the study results involved "quantitising" data. Quanititising is a common approach for enabling data transformation and comparison and can be defined as assigning a numerical value to data conceived as nonnumerical [56]. Assigning a descriptive strength grading to quantitative data in this analysis can also be considered a form of qualitising, converting quantitative data into qualitative data [56]. Qualitative

**Table 2. Example of the matrix for the mixed method analytical approach.** Illustrating the approach to devising a framework for integrating the evidence across the methods.

| Barrier | Study Method 1 | Study Method 2 | Study Method 3 | Etc. |
|---|---|---|---|---|
| A | Description of findings<br>Grade evidence as an important health system barrier (strong, moderate, weak, silent). | Description of findings<br>Grade evidence as an important health system barrier (strong, moderate, weak, silent). | Description of findings<br>Grade evidence as an important health system barrier (strong, moderate, weak, silent). | |
| B etc. | | | | |

**Table 3. Framework for grading strength of evidence in the mixed method cross-study analysis.** Qualitative narratives were also included within the matrix for community FGDs and photovoice. This allowed upgrading of the evidence grade when a narrative strongly suggested importance. Case-by-case decisions were taken where data were judged to not allow for a standardised approach for all barriers. Details of these decisions are evidenced in the matrix analysis results (S1 Table). Overall delay evidence gradings were similarly case by case.

| Study method | Delays and barriers evidenced | Strong Grade | Moderate Grade | Weak Grade | Silent Grade |
|---|---|---|---|---|---|
| Household survey | Overall delays 1 to 3, barriers only delay 1. | >10% of injured patients not seeking care report this barrier | 5% to 10% of injured patients not seeking care report this barrier | <5% of injured patients not seeking care report this barrier | Barrier not described |
| Verbal autopsy | Overall delays 1 to 3, barriers delays 1 to 3. | >20% of avoidable deaths within the delay evidence this barrier | 10% to 20% of avoidable deaths within the delay evidence this barrier | <10% of avoidable deaths within the delay evidence this barrier | Barrier not described |
| Community FGD | Overall delays 1 to 3, barriers delays 1 to 3. | All 3 FGDs describe this barrier | 2/3 FGDs describe this barrier. | 1/3 FGDs describe this barrier. | Barrier not described |
| Community photovoice | Overall delays 1 to 3, barriers delays 1 to 3. | 3 or more photos describe this barrier | 2 photos describe this barrier. | 1 photo describes this barrier. | Barrier not described |
| Facility process mapping | Overall delays 1 to 3, barriers delays 1 to 3. | Frequently reported, i.e., 6–11 facility maps evidence this barrier | Sometimes reported, i.e., 3–5 facility maps evidence this barrier | Rarely reported, i.e., 1–2 facility maps evidence this barrier | Barrier not described |
| Healthcare worker survey | Overall delays 1 to 3, barriers delays 1 to 3. | Either (a) within first 1/3 most important according to barrier score per delay or (b) within first 1/3 in top 3 overall barrier ranking (grade whichever is highest) | Either (a) within middle 1/3 most important according to barrier score per delay or (b) within middle 1/3 in top 3 overall barrier ranking (grade whichever is highest) | Either (a) within 1/3 least important according to barrier score per delay or (b) within last 1/3 in top 3 overall barrier ranking (grade whichever is highest) | Barrier not described |
| Facility assessment | Overall delay 3, some barriers in delay 3. | Case by case | Case by case | Case by case | Case by case |
| Clinical vignettes | Overall delay 3, some barriers in delay 3. | Case by case | Case by case | Case by case | Case by case |
| GIS analysis | Overall delay 2. | | | | |

FGD, focus group discussion; GIS, geographic information system.

narratives were also included within the grading framework. This allowed upgrading of the evidence grade when a narrative strongly suggested importance. This grading approach was also taken for the overall evidence for each delay. The facility assessment and clinical vignettes evidence strength grading was judged on a case-by-case basis (by JW and IE) since the data from these methodologies did not fit the standardised approach used for other barriers. As common with convergent parallel approaches, no greater weighting was given to one study method over another [52].

Two authors (JW and IE) subsequently independently tested the agreed scoring framework against 40% sample of barriers (each of the 3 delays overall summaries and 8 individual barriers) exposed by the study methods. A weighted Cohen's kappa test for agreement was performed between the 2 authors' gradings [57]. All areas of disagreement from this sample were discussed until agreement, and a common understanding was reached. One author (JW) then continued to grade the remaining study evidence within the matrix.

## Ethical considerations

All participants gave fully informed consent to participate. The Household Survey was approved by the Malawi National Health Sciences Research Committee (ref 19/07/2368) and

the UK MOD Research and Ethics Committee (ref 961/MODEC/19). The remainder of the methods were approved by the Malawi National Health Sciences Research Committee (ref 19/03/2263) and the UK MOD Research and Ethics Committee (ref 960/MODEC/19). Where identifiable images are used, participant consent was obtained, including those from the community photovoice method.

### Patient and participant involvement

For the community methods involving primary data collection (household survey, FGDs, and photovoice), a community sensitization meeting took place. Traditional community heads were invited to attend a meeting where all aspects of the community methods were explained, and questions answered allowing onward dissemination to the community. This is routine practice on the introduction of new studies within the MEIRU Karonga HDSS [38].

## Results

We identified 26 barriers to access timely quality injury care evidenced by at least 3 of the 9 study methods. There were 10 barriers at delay 1, 6 at delay 2, and 10 at delay 3, of which 20 were also found in the Delphi-generated framework. Cohen's kappa for agreement between authors on the grading of the strength of evidence for importance of barriers was 0.688 (95% confidence interval 0.525, 0.851), demonstrating good agreement [57].

The barriers, their description, and the corresponding strength of evidence that they are important within the studied health system are displayed in matrix form in summary in Table 4 and in detail in S1 Table. Within the table, matrix barriers are displayed in order according to the number of methods finding strong evidence that they are important. Those barriers with the greater number of methods finding strong evidence that they were important are at the top. Within delay 1, the barrier "cost" is at the top with 5 methods finding strong and 1 method moderate evidence that it is an important health system barrier. Within delay 2, the barrier "transport" is at the top with 5 methods (all those applicable) finding strong evidence that it is an important health system barrier. Within delay 3, the barrier "physical resources" is at the top with 7 methods (all those applicable) finding strong evidence that it is an important health system barrier.

Overall, 6 barriers within delay 1 had strong evidence that they were important in the majority of applicable methodologies. These were "cost" (5/6), "healthcare literacy" (4/6), "cultural norms" (4/6), "traditional healers" (3/6), "perceived physical access" (3/6), and "community or bystander engagement" (3/6). Within delay 2, 3 barriers had strong evidence that they were important in the majority of applicable methods, *"transport"* (5/5), *"roads"* (4/5), and *"distance"* (4/5). For delay 3, 5 barriers had strong evidence of being important in the majority of applicable methods, *"physical resources"* (7/7), *"staff"* (6/7), *"specialists"* (5/7), *"interfacility transfer"* (5/7), and *"quality processes"* (4/7).

Facility process mapping was the method that provided evidence for the greatest number of barriers, 25 out of 26 within the integrated analysis. This was followed by the healthcare worker survey (24 out of 26), community FGDs (22 out of 26), and verbal autopsy (18 out of 26).

There were some barriers with notable divergent findings between the community- and facility-based methods. This included the barrier "*cultural norms,*" which had only weak evidence of importance from community qualitative methods, only being reported in one FGD and not at all in the photovoice evidence, despite strong facility-based evidence. "*Community or bystander engagement*" had only weak evidence of importance from facility process mapping, featuring in 1/11 facility process maps and not raised in the healthcare worker survey, despite strong community-derived evidence. "*Alleged corruption*" was strongly evidenced in

**Table 4. Mixed method matrix summary of evidence strength grades according to barrier within each of the 3 delays.** Barriers are in order of evidence strength. Not all study methods could, by design, evidence all the delays (labelled as Not Applicable). Barriers for which a study methodology provided no evidence were categorised as silent. This silence could be a limitation of a method being unsuitable to detect evidence for a barrier, or suggestive that the barrier does not exist. A traffic light colour code, green for strong, amber for moderate, and red for weak has been used to facilitate interpretation.

| | Household Survey Evidence | Verbal Autopsy Evidence | Community FGD Evidence | Community Photovoice Evidence | Facility Process Mapping Evidence | Healthcare Worker Survey Evidence | GIS Evidence | Facility Assessment Evidence | Clinical Vignettes Evidence |
|---|---|---|---|---|---|---|---|---|---|
| Delay 1 Overall | Moderate | Strong | Strong | Strong | Strong | Strong | Not Applicable | Not Applicable | Not Applicable |
| Delay 1 Barriers | | | | | | | | | |
| COST—The financial costs associated with seeking care are too great | Strong | Moderate | Strong | Strong | Strong | Strong | Not Applicable | Not Applicable | Not Applicable |
| HEALTHCARE LITERACY-People do not understand about health and available healthcare | Strong | Strong | Strong | Silent | Strong | Moderate | Not Applicable | Not Applicable | Not Applicable |
| CULTURAL NORMS—Normal cultural behaviours delay seeking care such as gender roles, family responsibilities and requiring someone else's permission to seek care | Strong | Strong | Weak | Silent | Strong | Strong | Not Applicable | Not Applicable | Not Applicable |
| TRADITIONAL HEALERS-People prefer traditional healers | Moderate | Weak | Strong | Strong | Strong | Moderate | Not Applicable | Not Applicable | Not Applicable |
| PERCEIVED PHYSICAL ACCESS—People perceive that care is too difficult to physically access | Strong | Silent | Strong | Silent | Strong | Moderate | Not Applicable | Not Applicable | Not Applicable |
| COMMUNITY OR BYSTANDER ENGAGEMENT–Not enough is done by fellow citizens to support care-seeking | Strong | Silent | Strong | Strong | Weak | Silent | Not Applicable | Not Applicable | Not Applicable |
| PERCEIVED CARE QUALITY—People perceive that available facility care is poor quality | Moderate | Silent | Strong | Moderate | Strong | Moderate | Not Applicable | Not Applicable | Not Applicable |
| DELAYED DISCOVERY—There are delays in discovering injured people, including because of intoxication | Weak | Strong | Weak | Silent | Moderate | Weak | Not Applicable | Not Applicable | Not Applicable |
| RELIGIOUS OR OTHER BELIEFS–believing that seeking formal healthcare is itself wrong | Weak | Silent | Weak | Moderate | Strong | Silent | Not Applicable | Not Applicable | Not Applicable |
| FEAR OR LACKING COURAGE–irrational incapacitation or rational concern of consequences such as being accused | Weak | Silent | Moderate | Silent | Moderate | Weak | Not Applicable | Not Applicable | Not Applicable |
| Delay 2 Overall | Moderate | Strong | Strong | Strong | Strong | Strong | Moderate | Not Applicable | Not Applicable |
| Delay 2 Barriers | | | | | | | | | |
| TRANSPORT—There is a lack of timely affordable emergency transport (formal or informal) | Not Applicable | Strong | Strong | Strong | Strong | Strong | Not Applicable | Not Applicable | Not Applicable |
| ROADS—There is a lack of reliable uncongested roads with priority for emergency vehicles | Not Applicable | Strong | Strong | Strong | Strong | Weak | Not Applicable | Not Applicable | Not Applicable |

*(Continued)*

**Table 4.** (Continued)

| | Household Survey Evidence | Verbal Autopsy Evidence | Community FGD Evidence | Community Photovoice Evidence | Facility Process Mapping Evidence | Healthcare Worker Survey Evidence | GIS Evidence | Facility Assessment Evidence | Clinical Vignettes Evidence |
|---|---|---|---|---|---|---|---|---|---|
| DISTANCE—There is a large physical distance from place of injury to an appropriate healthcare facility | Not Applicable | Strong | Strong | Silent | Strong | Strong | Not Applicable | Not Applicable | Not Applicable |
| PREHOSPITAL CARE—There is a lack of timely available prehospital emergency care (formal or informal/bystander) | Not Applicable | Strong | Silent | Silent | Weak | Moderate | Not Applicable | Not Applicable | Not Applicable |
| COMMUNICATION—There is a lack of accessible emergency assistance communication mechanism (e.g., emergency call centre) | Not Applicable | Strong | Silent | Silent | Weak | Moderate | Not Applicable | Not Applicable | Not Applicable |
| COORDINATION—There is a lack of emergency care service coordination, including bypassing unsuitable facilities | Not Applicable | Strong | Silent | Silent | Weak | Weak | Not Applicable | Not Applicable | Not Applicable |
| Delay 3 Overall | Weak | Strong | Strong | Strong | Strong | Moderate | | Strong | Strong |
| Delay 3 Barriers | | | | | | | | | |
| PHYSICAL RESOURCES—There is a lack of reliably available necessary physical resources (e.g., infrastructure, equipment, and consumable material) | Not Applicable | Strong | Strong | Strong | Strong | Strong | Not Applicable | Strong | Strong |
| STAFF—In regard to staffing, there is a lack of reliably available, suitably trained, and motivated clinical staff | Not Applicable | Strong | Strong | Moderate | Strong | Strong | Not Applicable | Strong | Strong |
| SPECIALISTS—There is a lack of reliable, timely access to specialist injury care services | Not Applicable | Strong | Strong | Moderate | Weak | Strong | Not Applicable | Strong | Strong |
| INTERFACILITY TRANSFER—Lack of available means to safely and quickly transfer injured patients on to a more specialist facility. | Not Applicable | Strong | Strong | Strong | Strong | Strong | Not Applicable | Moderate | Silent |
| QUALITY PROCESSES—There is a lack of good quality, consistent, structured, clinical priority-driven injury care processes | Not Applicable | Strong | Strong | Moderate | Moderate | Moderate | Not Applicable | Strong | Strong |
| ALLEGED CORRUPTION—Need for unauthorised payments or gifts to healthcare staff to receive best available treatment (e.g., corruption) | Not Applicable | Silent | Strong | Strong | Silent | Weak | Not Applicable | Silent | Silent |
| POLICE PROCESSES–perceived or actual police functions affect care access | Not Applicable | Weak | Moderate | Silent | Strong | Weak | Not Applicable | Silent | Silent |
| PAYMENT—Difficulties with timely payment for care | Not Applicable | Weak | Moderate | Moderate | Weak | Weak | Not Applicable | Silent | Silent |

(*Continued*)

**Table 4.** (Continued)

| | Household Survey Evidence | Verbal Autopsy Evidence | Community FGD Evidence | Community Photovoice Evidence | Facility Process Mapping Evidence | Healthcare Worker Survey Evidence | GIS Evidence | Facility Assessment Evidence | Clinical Vignettes Evidence |
|---|---|---|---|---|---|---|---|---|---|
| PATIENT COOPERATION—There is a lack of patient and family cooperation with care processes | Not Applicable | Silent | Moderate | Silent | Weak | Moderate | Not Applicable | Silent | Silent |
| CAPACITY—In regard to patient demand, there is insufficient facility capacity to meet patient demand (e.g., overcrowding) | Not Applicable | Silent | Silent | Silent | Moderate | Moderate | Not Applicable | Silent | Silent |

community qualitative methods (FGDs and photovoice) while only weakly evidenced in the healthcare worker survey and no evidence from facility process mapping, facility assessment, or clinical vignettes.

There was some divergence of findings among the facility-based methods. The barriers "*roads*" and "*police processes*" were strongly evidenced as important in facility process mapping but only weakly in the healthcare worker survey. "*Religious or other beliefs*" were also strongly evidenced as important through facility process mapping but not at all in the healthcare worker survey. Conversely, the barrier "*specialists*" had strong evidence of importance from the healthcare worker survey but was not prioritised in facility process mapping.

Among community-based methods, there was also some observed divergence. The delay 1 barriers "*healthcare literacy*" and "*perceived physical access*" were both strongly evidenced in the household survey and FGDs but not evidenced by the photovoice method. Similarly, the barriers "*distance,*" "*police processes,*" and "*patient cooperation*" were strongly or moderately evidenced as important by FGDs but not evidenced in the photovoice method.

There was other notable divergence between some of the study methods in evidencing barriers. The delay 2 (reaching care) barriers, "*prehospital care,*" "*communication,*" and "*coordination,*" were strongly evidenced by the verbal autopsy method. Conversely, there was no evidence from the qualitative community methods, and only weak or moderate evidence from the facility-based methods. This contrasted with delay 1 (seeking care), where several barriers including "*perceived physical access,*" "*perceived care quality,*" "*community or bystander engagement,*" and "*religious or other beliefs*" had strong evidence of importance from other methods that were not evidenced by the verbal autopsy method. The same was found for the delay 3 (receiving care) barrier "*alleged corruption.*"

## Discussion

Our study used multiple methods to holistically capture important barriers to seeking, reaching, and receiving timely quality care following injury. Using a, to the best of our knowledge, novel consensus approach, we graded the evidence from each method as to whether an identified barrier was important within the health system. This grading facilitated integration of the results from individual methods to reveal those barriers with the most methods providing strong evidence that they were an important health system barrier. We used a tabulated matrix to order identified barriers according to the number of methods providing strong evidence they are an important health system barrier. We found that the barriers "cost," "transport," and "physical resources" had the most methods providing strong evidence they were important health system barriers within delays 1, 2, and 3, respectively, suggesting the importance of

prioritising these barriers in order to improve the health system's ability to care for people who have been injured.

Integrating findings from 9 diverse methods allowed a deeper understanding of the complexity of the health system for trauma care, viewed from multiple different lenses. It represents a valuable approach in this field where most previous studies assessing barriers to access to care following injury have focused on a single aspect of the health system [8,58]. Most approaches to assessing barriers have also been single method approaches, which can fail to capture both community and facility evidence concurrently [8]. Other attempts to combine methods to understand barriers to injury care have shown, as we have, that using multiple methods identifies important barriers that would have been missed through using single methods alone [59]. Attempts to prioritise the importance of injury care barriers in health systems have been done using Delphi methods [30] and workshop ranking exercises [4,59,60]. However, we used an integrative analytical approach combining qualitising and quantitising techniques to present findings through a joint matrix display, bringing the various data together through a visual format. This process highlighted and ordered barriers with the most methods providing strong evidence they are important for application to health system strengthening, thereby providing insights beyond a siloed analysis of results [52,61]. We found the use of Strong, Moderate, Weak, and Silent data labels in our comparative matrix valuable to supporting integration in our study and propose this as an exemplar for wider application in mixed method health system research [62,63]. Convergent study design joint displays are often themes-by-statistics or side-by-side comparisons; however, due to the unusually large number of different methods being compared in our study, we believe we have showed novel depth to the joint display integration [62] to support quicker interpretation of results [64].

Noteworthy divergent findings within mixed method analysis should be welcomed. They can provide opportunity for creative insight into the methods used or the study context, stimulate interpretive explanations, and set future research priorities [20]. They can also shape choice of future methods for similar assessments to ensure all barriers are evaluated. Divergence of findings between community- and facility-based methods in our study included the barriers "*cultural norms*" with weak evidence from community qualitative methods in contrast to strong facility-based evidence. Community social and cultural perceptions and gender-based factors are known barriers to women seeking care [4,22,30,65,66]. Male household heads can have higher disease severity thresholds for financing health-seeking, more readily financing healthcare for male household members [67]. All of our FGD groups were mixed sex. Such issues might be more likely to arise in single-sex discussion groups [68] as a sensitive topic that only community leaders felt empowered to raise. Community photovoice and FGD participants were not provided with the Delphi [30] derived barriers but generated ideas themselves without opportunity to confirm (or deny) this phenomenon specifically. Future qualitative enquiry could perhaps better explore this issue directly.

"*Alleged corruption*" was strongly evidenced in community qualitative methods (FGDs and photovoice), while only weakly evidenced in the healthcare worker survey, with no evidence from facility process mapping, facility assessment, or clinical vignettes. Differences between community and facility staff alleging client differentiation based on personal connections or money are common in underresourced health systems [69]. Insights from healthcare workers on this subject may be better obtained through specifically directed discussions [70] rather than the more open approaches we and others have used [69]. Since such practices may have legal or disciplinary implications, a reluctance to discuss them from healthcare workers is understandable [71]. The use of qualitative methods directed to understand this complex and sensitive subject is potentially more suitable and may suspend preconceptions about the controversial subject [71].

Divergence of findings was also seen between facility-based methods. The barrier "*religious or other beliefs*" and "*police processes*" were strongly evidenced in facility process mapping but only weakly or not at all in the healthcare worker survey. The healthcare worker survey asked directly about a priori Delphi study derived conceptual barriers [30] and offered free text response for participants to propose new ideas. Survey free text responses, commonly used to "safety net" against missing themes [72], were rarely novel and provided by only a minority of participants, as is common for usually unclear reasons [73,74]. Process mapping is interactive and engaging [75] supporting idea generation, particularly from those who understand systems, but infrequently contribute to system-level research and innovation [75]. It also helps "brainstorming" to recall all important aspects, or barriers, in a process [76]. Deep participant engagement for new insights may be less obtainable through surveys than the exercise of process mapping.

The delay 2 barriers "*prehospital care*," "*communication*," and "*coordination*" were strongly evidenced as important by the verbal autopsy method. There was conversely no evidence from the qualitative community methods and only weak or moderate evidence of importance from the facility-based methods. This could reflect the local context of nonexistent formal prehospital emergency care [77,78], unlike other sub-Saharan African contexts where such systems are rapidly developing [79]. Participants cannot want something they do not have prior knowledge of and are therefore unlikely to bring up such issues for discussion without specific prompting. The verbal autopsy analysis was conducted by researchers with an awareness of the potential benefits of formal mature prehospital care systems, who brought this knowledge to the analysis of potential avoidable deaths [5].

Reflecting on our chosen methods, we found process mapping to be highly practical, easy to use, and quickly taught and performed [80], succinctly framing complex health systems problems [81–83]. It evidenced the largest numbers of barriers within our analysis, perhaps as it encouraged participants to consider specific differences between injury mechanisms and contexts. As with any group-based method, workshop discussions may be tempered by hierarchical power dynamics [84,85], perhaps leading to disproportionate contributions from clinicians, especially more senior clinicians, compared to nonclinical staff. Since we used process mapping in facilities with healthcare workers as participants, patient or community perspectives were missing. However, incorporating community members would have required local language facilitation with pragmatic research conduct implications.

The community qualitative, household survey, and verbal autopsy methods provided empirical population-derived data for the integrated analysis. Verbal autopsy data were available because of an established longitudinal surveillance programme in this population [38,86]. While verbal autopsy is a valuable potential source of community-derived insight, its broader application to other populations without existing demographic surveillance is limited. Our household survey allowed access to quantitative community data about nonfatal injuries, including those not presenting for healthcare [87]. It was, however, the most expensive method to undertake, approximating to the combined costs of the other methods. This was due to the large resource requirements to administer in-person surveys in a context where internet or postal surveys are impractical. Community qualitative studies, such as FGDs, can potentially generate insight into health system barriers in a short time frame and be readily replicated in other contexts [88,89]. While we incorporated a photovoice extension to one focus group, the additional time, resources, and ethical considerations need to be balanced with the benefits of potential additional insight and further value of pictural representation of barriers of interest [90].

Considering evaluating delay 2, the availability of powerful computing, along with large-scale and open-source data, has led to growth in GIS methods in health system research [48].

GIS has many attractive features. Various open and freely accessible data sources can be used to provide a theoretical spatial access dimension to the subject of interest [91,92]. However, attempts at validation with patient data have shown they can provide overly optimistic estimates of patient reported time to reach urgent care facilities. Therefore, care must be taken with interpretation of GIS models, which may need to be appropriately adjusted to better reflect population geospatial care access [91,93–96]. Furthermore, access to care is a multifaceted concept including additional dimensions such as cost, cultural norms, and health system responsiveness [4,5,97]. While the temporal dimension to care access that GIS helps estimate is a fundamental component of care access [98], assessing this together with all the other factors influencing emergency care access necessitates integration with other methods. Our mixed method approach has illustrated how this can be done.

Surveys for structures of care are common across LMIC health systems research. We found, as have others, that facility assessments can be quick to complete [99]. They can be repeated and allow comparison across settings but are not suited to assess either delay 1 or 2 or process and outcome dimensions of care [100]. Outcome measures of care can require longitudinal data, such as verbal autopsy in our study, or trauma registries more commonly. Trauma registries remain limited in their uptake in LMICs and only present in a few locations in Malawi [36,37,101,102]. To complement the facility assessment, we used clinical vignettes to assess provider care quality [45,46,96]. Vignettes are frugal relative to care observation and case note review [103], both being constrained by feasibility, sample size, and reporting accuracy [104]. However, vignettes can facilitate standardised comparison of clinicians across facilities [103] and offer a pragmatic method for assessing provider care quality. Vignettes are low cost compared to alternatives such as observation [105]. They also do not depend on either the providers' routine workload nor on patient selection [106,107], and we found vignettes could be undertaken fairly quickly, around 30 minutes per clinician.

In the relatively underresourced and underresearched field of trauma in LMICs [108], efficiency in health system assessments is needed to identify needs and monitor interventions [109,110]. Rapid assessment principles appraise the potential value of methods' suitability for inclusion in efficient and holistic mixed-method health system assessment [111]. These principles are cost-effectiveness, speed, use of multiple data sources, and pragmatism [111]. Cost and speed are intersecting principles since additional resources may speed up assessment processes. Pragmatic metrics are achievable, not arduous to collect, readily interpretable, and widely applicable [112], providing sufficient rather than perfect information [111]. Triangulating findings from multiple data sources to evaluate convergence or divergence of findings is an inherent feature of mixed methods research [52].

Summarised in Table 5, with more detailed explanation in S2 Table, we include our reflections on how the methods in this study perform in line with these principles and their potential for adaptation for future use to improve compliance with these principles of efficient health system assessment. Our reflections are summarised in a grading of well aligned, partly aligned, or poorly aligned.

According to these reflections, process mapping was the most compliant method, scoring as well-aligned with all 4 principles for both performance and potential. We would strongly advocate for its wider adoption by others for including in mixed-method health system assessment. Community FGDs similarly performed well, and we would recommend to include this in assessments also. Pragmatic and cost considerations around administration resource requirements constrain our recommendations for inclusion of both household and health care worker surveys. Similarly, challenges in triangulating and integrating the findings from GIS, verbal autopsy, and clinical vignettes should be carefully considered before including in future mixed-method health system assessments. That being said, the specific purpose for a health

**Table 5. Summary of reflections on the compliance of study methods with the rapid assessment principles (authors' judgement).** Performance refers to authors' experience, and potential refers to the scope for adaptation to improve compliance with a rapid assessment principle. These represent the authors' reflections. Well-aligned (green), partly well-aligned (yellow), or poorly aligned (red).

| Method | | Speed | Pragmatism | Cost-effectiveness | Triangulation |
|---|---|---|---|---|---|
| Household survey | Performance | Partly well-aligned | Partly well-aligned | Poorly aligned | Partly well-aligned |
| | Potential | Well-aligned | Partly well-aligned | Poorly aligned | Well-aligned |
| Verbal autopsy | Performance | Well-aligned | Well-aligned | Well-aligned | Poorly aligned |
| | Potential | Well-aligned | Poorly aligned | Well-aligned | Poorly aligned |
| FGDs | Performance | Partly well-aligned | Well-aligned | Well-aligned | Well-aligned |
| | Potential | Well-aligned | Well-aligned | Well-aligned | Well-aligned |
| Community photovoice | Performance | Well-aligned | Partly well-aligned | Well-aligned | Partly well-aligned |
| | Potential | Well-aligned | Well-aligned | Well-aligned | Well-aligned |
| Process mapping | Performance | Well-aligned | Well-aligned | Well-aligned | Well-aligned |
| | Potential | Well-aligned | Well-aligned | Well-aligned | Well-aligned |
| Healthcare worker survey | Performance | Poorly aligned | Partly well-aligned | Partly well-aligned | Well-aligned |
| | Potential | Well-aligned | Partly well-aligned | Well-aligned | Well-aligned |
| GIS analysis | Performance | Well-aligned | Well-aligned | Well-aligned | Partly well-aligned |
| | Potential | Well-aligned | Well-aligned | Well-aligned | Partly well-aligned |
| Clinical vignettes | Performance | Partly well-aligned | Well-aligned | Well-aligned | Partly well-aligned |
| | Potential | Well-aligned | Well-aligned | Well-aligned | Partly well-aligned |
| Facility assessment | Performance | Well-aligned | Well-aligned | Well-aligned | Partly well-aligned |
| | Potential | Well-aligned | Well-aligned | Well-aligned | Well-aligned |

FGD, focus group discussion; GIS, geographic information system.

system assessment should nuance the choice of methods included. For example, including photovoice, while perhaps not adding substantially to information about barriers to care in this study, could provide images that may be useful for advocating for intervention [113]. Similarly, if methods, such as process mapping, are planned to take place within a facility, including a facility assessment may be justifiable at the same visit without adding substantially to the cost or time required for the study.

We propose, therefore, that the choice of methods used for future holistic health system assessment should balance (a) the pragmatism of encompassing suitable preexisting data sources, such as verbal autopsy and GIS in our study or trauma registry data where available; (b) the economy of low cost and practical methods such as community focus groups, facility process mapping, and facility assessments; and (c) the resource requirements of methods such as vignettes, or especially administered surveys to either households or healthcare workers. The choices will depend to some degree on specific health system research aims, local research infrastructure and data, and resources available for a given project.

There are limitations to this study. Since we chose a convergent parallel design, a more in-depth exploration of some of the observed divergent findings might have been possible through a sequential design. For example, in an explanatory format, targeting FGDs towards results of the surveys, or an exploratory format, designing the survey questions after community qualitative and facility process mapping methods had been analysed. However, sequential approaches would likely be more time consuming, costly, and still have potential to miss important concepts. The integrated matrix analysis focused on the comparison of barriers within each delay, rather than across all delays, which could have been alternatively explored. Our framework for grading evidence strength for each barrier might have had different thresholds if different individuals were constructing it. Similarly, the kappa for agreement could have

been stronger. However, our broad approach to devise and undertake a coherent analytical approach to meaningfully compare multiple different study methods remains transferable, and we would encourage other researchers to further explore and test frameworks for meaningful cross-method integration and interpretation. We used an analytical strategy that provided equal weight to each method. Similarly, equal weighting was given to participants in each method. It might have been possible to provide hierarchical bias to responses, for example, in the FGDs, injured participants may have been considered to have more insight into injury care quality, or the community leaders may have been considered more authoritative on issues related to community engagement and behaviour. Likewise, facility process maps were a group-derived output, which, although potentially influenced by personality, cultural and hierarchical roles, were not weighted to value the clinician's views more than those of nonclinicians. As well, this study was limited to a rural population. Urban and rural populations in sub-Saharan Africa experience differing burdens of injury [114]. The scientific value of dichotomising rural and urban contexts has been challenged for some health-related questions [115]. However, the health system function and barriers for accessing high-quality care would still be a valuable frame for such a comparison. Future application of this approach could include urban population comparisons to incorporate any differences. Finally, what was found to be important in Malawi may differ from other contexts, which may, in turn, influence the methodologies required to identify them.

The potential value of mixed methods in emergency health systems research is recognised, but its use is in its infancy [58]. True mixed-methods studies integrating data before drawing conclusions in trauma care system assessments are rare [8]. Our study, therefore, makes an important contribution to the existing literature in the application of mixed methods approaches to health system research in general, and trauma care system research in particular. We are not aware of any similar in-depth assessment on a single population's health system using as many as 9 different methods. Our study, to our knowledge, represents a novel approach in both the weight and focus of a mixed-method study. We found the barriers of cost, transportation, and physical resources were most strongly evidenced across the methods. While previous single method studies have highlighted these health system issues [8], our study has found them to be the most important barriers in the health system that we assessed and should end further speculation on their relative contribution to delays in accessing quality care. We hope this encourages other researchers to use similar approaches to holistically assess health systems in other contexts. Drawing on this study's experience, funding has been granted for further holistic mixed method trauma health system assessment across 4 LMICs [116].

## Supporting information

**S1 GRAMMS Checklist. Good Reporting of A Mixed Methods Study (GRAMMS checklist).** (DOCX)

**S1 Text. Unpublished methods supporting information.** (DOCX)

**S2 Text. Inclusivity in global research questionnaire.** (DOCX)

**S1 Table. Mixed method integrated analysis detail of evidence grading for each study, for each of the 3 delays and identified barriers.** (DOCX)

**S2 Table. Detail of reflections on the compliance of study methods with the rapid assessment principles (authors' judgement).**
(DOCX)

## Author Contributions

**Conceptualization:** John Whitaker, Idara Edem, Rory Rickard, Andrew JM Leather, Justine Davies.

**Data curation:** John Whitaker, Ella Togun.

**Formal analysis:** John Whitaker, Idara Edem, Ella Togun, Albert Dube, Giulia Brunelli, Thomas Van Boeckel.

**Funding acquisition:** John Whitaker, Rory Rickard, Andrew JM Leather, Justine Davies.

**Investigation:** John Whitaker, Ella Togun, Abena S. Amoah, Albert Dube.

**Methodology:** John Whitaker, Albert Dube, Andrew JM Leather, Justine Davies.

**Project administration:** John Whitaker, Ella Togun, Abena S. Amoah, Lindani Chirwa, Boston Munthali.

**Resources:** John Whitaker, Abena S. Amoah, Lindani Chirwa, Boston Munthali, Andrew JM Leather, Justine Davies.

**Supervision:** Abena S. Amoah, Albert Dube, Rory Rickard, Andrew JM Leather, Justine Davies.

**Validation:** John Whitaker, Idara Edem, Albert Dube.

**Visualization:** John Whitaker.

**Writing – original draft:** John Whitaker.

**Writing – review & editing:** John Whitaker, Idara Edem, Ella Togun, Abena S. Amoah, Albert Dube, Lindani Chirwa, Boston Munthali, Giulia Brunelli, Thomas Van Boeckel, Rory Rickard, Andrew JM Leather, Justine Davies.

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
