## [Editor Report · Decision Letter 0]

7 Nov 2022

Dear Dr Whitaker, 

Thank you for submitting your manuscript entitled "A holistic health system assessment for access to care after injury in low- or middle-income countries: an exemplar study from Northern Malawi." for consideration by PLOS Medicine.

Your manuscript has now been evaluated by the PLOS Medicine editorial staff as well as by an academic editor with relevant expertise and I am writing to let you know that we would like to send your submission out for external peer review.

Please re-submit your manuscript within two working days, i.e. by Nov 09 2022 11:59PM.

Kind regards,

Callam Davidson

Senior Editor

PLOS Medicine

---

## [Decision Letter · Decision Letter 1]

5 Jul 2023

Dear Dr. Whitaker,

Thank you very much for submitting your manuscript "A holistic health system assessment for access to care after injury in low- or middle-income countries: an exemplar study from Northern Malawi." (PMEDICINE-D-22-03524R1) for consideration at PLOS Medicine. 

Your paper was evaluated by an associate editor and discussed among all the editors here. It was also discussed with an academic editor with relevant expertise, and sent to independent reviewers, including a statistical reviewer. The reviews are appended at the bottom of this email and any accompanying reviewer attachments can be seen via the link below:

[LINK]

In light of these reviews, I am afraid that we will not be able to accept the manuscript for publication in the journal in its current form, but we would like to consider a revised version that addresses the reviewers' and editors' comments. Obviously we cannot make any decision about publication until we have seen the revised manuscript and your response, and we plan to seek re-review by one or more of the reviewers. 

We expect to receive your revised manuscript by Jul 24 2023 11:59PM. Please email us (plosmedicine@plos.org) if you have any questions or concerns.

We look forward to receiving your revised manuscript. 

Sincerely,

Alexandra Schaefer, PhD

PLOS Medicine

plosmedicine.org

GENERAL COMMENTS

Please respond to all editor and reviewer comments. 

Please include page numbers and line numbers in the manuscript file. Use continuous line numbers (do not restart the numbering on each page). Only for review purposes, the editorial team decided to number the pages of your manuscript starting with the abstract page as 1.

Please cite the reference numbers in square brackets (e.g., “We used the techniques developed by our colleagues [19] to analyze the data”). Citations should be preceding punctuation.

Please cite your Supporting Information as outlined here: https://journals.plos.org/plosmedicine/s/supporting-information. 

Please ensure to be consistent in the use of numbers as words or numerals.

Please ensure to be consistent in writing ‘mixed-methods’ or ‘mixed methods’.

Please report your qualitative study according to the appropriate study design provided at http://www.equator-network.org/?post_type=eq_guidelines&eq_guidelines_study_design=qualitative-research&eq_guidelines_clinical_specialty=0&eq_guidelines_report_section=0&s= and provide the relevant completed checklist. In general, we recommend that authors use the COREQ checklist, or other relevant checklists listed by the Equator Network, such as the SRQR, to ensure complete reporting (http://journals.plos.org/plosone/s/submission-guidelines#loc-qualitative-research). In the checklist, please include sufficient text excerpted from the manuscript to explain how you accomplished all applicable items. 

Please describe the conceptual framework underlying your qualitative analysis.

In general, we expect qualitative studies to include the following: 1) defined objectives or research questions; 2) description of the sampling strategy, including rationale for the recruitment method, participant inclusion/exclusion criteria and the number of participants recruited; 3) detailed reporting of the data collection procedures; 4) data analysis procedures described in sufficient detail to enable replication; 5) a discussion of potential sources of bias; and 6) a discussion of limitations.

ACADEMIC EDITOR COMMENTS

This is an innovative paper in an important area that could benefit however from the very nice set of suggestions made by reviewer 2.

COMPETING INTEREST STATEMENT

All authors must declare their relevant competing interests per the PLOS policy, which can be seen here: https://journals.plos.org/plosmedicine/s/competing-interests

For authors with ties to industry, please indicate whether any of the interests has a financial stake in the results of the current study.

ABSTRACT

Please ensure that all numbers presented in the abstract are present and identical to numbers presented in the main manuscript text.

PLOS Medicine requests that main results are quantified with 95% CIs as well as p values. Please include. When reporting p values please report as p<0.001 and where higher as the exact p value p=0.002, for example. For the purposes of transparent data reporting, if not including the aforementioned please clearly state the reasons why not.

Please include any important dependent variables that are adjusted for in the analyses.

Throughout, suggest reporting statistical information as follows to improve clarity for the reader “22% (95% CI [13%,28%]; p</=)”. Please amend throughout the abstract and main manuscript.

Please note the use of commas to separate upper and lower bounds, as opposed to hyphens as these can be confused with reporting of negative values.

When a p value is given, please specify the statistical test used to determine it. Please report p values as p<0.001 and where higher as 'p=0.002'

Abstract Methods and Findings:

* Please ensure that all numbers presented in the abstract are present and identical to numbers presented in the main manuscript text.

* Please include the study design, population and setting, number of participants, years during which the study took place, length of follow up, and main outcome measures.

Abstract Conclusions:

* Please avoid vague statements such as "these results have major implications for policy/clinical care". Mention only specific implications substantiated by the results.

* Please temper assertions of primacy ("Our novel approach to integrating […]”) by adding ‘to the best of our knowledge’ or similar.

p.1: Please change ‘haven’t’ to ‘have not’ and refrain from using contractions.

p.1: Please change ‘Delay’ to ‘delay’.

p.1: Please change ‘Three Delays’ to ‘three delays’.

p.1: Please define ‘HDSS’.

AUTHOR SUMMARY

Thank you for providing a short, non-technical Author Summary of your research to make findings accessible to a wide audience that includes both scientists and non-scientists. This text is subject to editorial change and should be distinct from the scientific abstract. The Author Summary should immediately follow the Abstract in your revised manuscript. Please see our author guidelines for more information: https://journals.plos.org/plosmedicine/s/revising-your-manuscript#loc-author-summary.

The summary should include 2-3 single sentence, individual bullet points under each of the questions.

It may be helpful to review currently published articles for examples which can be found on our website here https://journals.plos.org/plosmedicine/

INTRODUCTION

Please address past research and explain the need for and potential importance of your study. Indicate whether your study is novel and how you determined that. If there has been a systematic review of the evidence related to your study (or you have conducted one), please refer to and reference that review and indicate whether it supports the need for your study. 

METHODS AND RESULTS

PLOS Medicine requests that main results are quantified with 95% CIs as well as p values. Please include. When reporting p values please report as p<0.001 and where higher as the exact p value p=0.002, for example. For the purposes of transparent data reporting, if not including the aforementioned please clearly state the reasons why not.

Please include any important dependent variables that are adjusted for in the analyses.

Suggest reporting statistical information as detailed above – see under ABSTRACT

Please present numerators and denominators for percentages, at least in the Tables [not necessarily each time they're mentioned].

In the Methods section, be sure to refer to the Supplementary Material when introducing the different methods used to examine the health care system for injuries (Methods, Study Setting, paragraph 4).

p.4 suggest: “Malawi is a landlocked, low-income country; the 6th poorest in the world [32]. Half of its 19 million people are children, and 50% of the population lives below the poverty line [32].”

p.4 suggest: “It focused specifically on the health system serving the population of Karonga HDSS in Karonga District, Northern Malawi [37].”

p.4 suggest: “The HDSS population is mostly rural, although about 15% live in semi-urban settlements. Half of the population lives within one kilometer of a paved road [37].”

p.4. suggest: “Secondary care facilities include a government facility 70 km north and a CHAM facility 40 km south over difficult hilly terrain. Tertiary care is provided at a government facility in the regional capital, Mzuzu, 150 km south.”

p.5: Please change ‘Feb’ to ‘February’.

p.5: Please cite the specific supplementary material that you are referring to (“A more detailed description of the study methods are reported in supplementary material.”)

p.5: Please define ‘GRAMMS’.

p.6 suggest: “All areas of disagreement from this sample were discussed until agreement and a common understanding was reached.”

p.7: Pleasure ensure to be consistent in the use of the word ‘Delay’ (uppercade D) referring to the Three Delays or ‘delay’ in its original meaning. Please revise throughout your entire manuscript.

p.7 suggest: “Facility process mapping provided evidence for the greatest number of barriers, 25 out of 26 within the integrated analysis.”

p.7 suggest: “Six barriers within Delay 1 had strong evidence that they were important in the majority of applicable methodologies.”

p.7 suggest: “Within Delay 2, three barriers had strong evidence that they were important in the majority of applicable methods, […]”.

p.7 suggest: “For Delay 3, five barriers had strong evidence of being important in the majority of applicable methods, […]”.

DISCUSSION

Please present and organize the Discussion as follows: a short, clear summary of the article's findings; what the study adds to existing research and where and why the results may differ from previous research; strengths and limitations of the study; implications and next steps for research, clinical practice, and/or public policy; one-paragraph conclusion. Please remove any subheadings.

p.11: “This is particularly the case at peripheral facilities [6, 76, 77, 80], which are routinely, as

we found, the first port of call for initial management of the injured.” – Did you find this in this study? Otherwise, please add a citation or reference for the second part of this statement.

p.13: Please temper assertions of primacy ("Our novel approach to integrating […]”) by adding ‘to the best of our knowledge’ or similar.

TABLES

Please note the use of commas to separate upper and lower bounds, as opposed to hyphens as these can be confused with reporting of negative values. Suggest reporting statistical information as detailed above – see under ABSTRACT

Please provide titles and legends for all tables (including those in Supporting Information files).

Please define all abbreviations used in the table below each table (including those in Supporting Information files).

Please consider avoiding the use of red and green in order to make your figure more accessible to those with colour blindness.

Table 1: Please define ‘HDSS’, ‘FGD’, ‘N/A’, ‘GIS’.

Table 3: Please define ‘FGD’, ‘GIS’.

Table 3: Please include a specific reference for “Details of these decisions are evidenced in the matrix analysis results table.”.

Table 4: Please define ‘N/A’. For clarity, please define the meaning of ‘Silent’. 

Appendix 1 - Please define ‘FGD’, ‘HH’, ‘HCW’, ‘VA’, ‘D1’, ‘PV’, ‘GIS’, ‘DSS’, ‘CRH’, ‘KDH (main gov 2’)’, ‘MCH (gov 3’)’, ‘D2’, ‘WHO ETC’, ‘GP’, ‘COs', ‘MAs', ‘D3’, ‘KDH’, ‘NB’

Appendix 1 – Please define ‘Silent’ as grade. We noted the use of different terms (‘Not able to assess’, ‘No evidence found’, ‘No evidence revealed’, ‘Not discussed’), please ensure to define these and if applicable, use consistent expressions. 

SUPPLEMENTARY MATERIAL

Please define all abbreviations used in the supplementary material.

p.1: Please define ‘HDSS’, ‘WHO’.

p.2: Please define ‘MEIRU’.

p.3: Please define ‘IV’.

p.4: Please define ‘CHAM’.

p.5: Please define ‘CI’.

p.6: Please define ‘VA’. Please change ‘Sep’ to ‘September’.

p.7: Please define ‘LMIC’.

p.9: Please define ‘myself’ in “non-native speakers (myself and a female Nigerian).” We suggest using initials instead. 

p.13. Please change ‘72’ to ’72 hours’.

p.17: Please change ‘Jul – Oct 2019’ to ‘July – October 2019’. Please define ‘HCW’, ‘WHO ETC’.

Table S3: Are the numbers shown in some of the columns actual numbers from the study, or are they just examples? Please clarify.

Table S3: Please define ‘WHO’. Please change “It should take no more that 1 working day to

complete, depending on staff availability.” to “It should take no more than 1 working day to

complete, depending on staff availability.”.

Table S3: Please add headers to the different columns and explain when a field was left empty (e.g., ‘Numeric=’),

Table S3: Please define ‘HIV, ‘Hep B/C’, ‘°C’, ‘Kasi’, ‘CT’, ‘MRI’

REFERENCES

PLOS uses the numbered citation (citation-sequence) method and first six authors, et al.

Please ensure that journal name abbreviations match those found in the National Center for Biotechnology Information (NCBI) databases (http://www.ncbi.nlm.nih.gov/nlmcatalog/journals), and are appropriately formatted and capitalised.

Please also see https://journals.plos.org/plosmedicine/s/submission-guidelines#loc-references for further details on reference formatting. 

Where website addresses are cited, please specify the date of access. 

PARACHUTE RESEARCH 

We note that you conducted research or obtained samples in a foreign country. Did you consider including a local author as first or last author? If not, we recommend that you consider doing so in line with ICMJE's authorship requirements (https://www.icmje.org/recommendations/browse/roles-and-responsibilities/defining-the-role-of-authors-and-contributors.html). PLOS has a parachute research policy which aims to promote collaboration and inclusivity in global health research. You are required to complete PLOS’ questionnaire on inclusivity in global research and submit it with your revised paper. The policy and questionnaire can be found at https://journals.plos.org/plosone/s/best-practices-in-research-reporting.

Comments from the reviewers:

Reviewer #1: I commend the authors for their novel and important work. The idea of providing multiple perspectives on each conceptual delay is useful, but having novel method to 'grade' them is an interesting approach, and offers practical value. Overall this study offers a novel and holistic approach in assessing the three delays concept in trauma system in LMICs, and this could be useful in guiding further research or aid in implementation science. 

Abstract: The abstract is satisfactory.

Introduction: This is well written and explains the context and rationale of the study. 

Methods: This is well described, however having subheadings for this section will be helpful to break down the steps involved in data gathering, framework development, quantitisation, grading, scoring etc. Why were the nine methods chosen? It is also not quite clear what each method entails, perhaps adding a table to describe each method would be useful for clinicians readers to understand. 

Results: The results are well illustrated in the tables. The results section reads rather flat, significant results or results of importance are not highlighted or made clear. Perhaps a paragraph highlighting the barriers supported by the strongest evidence across multiple methods would be useful. 

Discussion: An interpretation of why the results are noteworthy or substantial need to be highlighted. The discussion doesn't quite explain why this study is significant, and what the results add to the current literature. Overall, I believe the discussion can be made more concise in several areas. 

I would suggest adding the below reference in discussing aspects in the third delay, especially regarding methods in improving the quality of care. Jin et al, Effectiveness of quality improvement processes, interventions, and structure in trauma systems in low-and middle-income countries: a systematic review and meta-analysis. World journal of surgery. 2021 Jul;45(7):1982-98. 

Reviewer #2: Thank you for the opportunity to review the study entitled "A holistic health system assessment for access to care after injury in low- or middle- income countries: an exemplar study from Northern Malawi." In this study, the authors implement a monumental and meticulous research methodology to analyze the trauma care system in Northern Malawi. They utilize previous literature and expert opinion to perform 9 separate studies to assess barriers to care. They identified 26 barriers present within their health system and discuss which methods were able to recognize which barriers. Overall, I found this project to be a significant contributor to the literature and believe it should be accepted after some revisions. 

My primary criticism of the paper is with the clarity in which the information is presented. This is in part related to the robustness of the study itself. While I believe this is a major strength of the project, it was difficult to keep track of what exactly was done for each of the 9 methodologies with 17 pages of supplemental material. The completeness in which this project was performed lead to an overview of several topics without proper depth of key findings. I will leave this to the publishers to decide but this project may best be presented as a series of studies in some other format rather than a single research publication.

As it stands, I do believe several changes can be made. 

* It is unclear whether the primary objective of the paper is to investigate barriers to care in Karonga, Northern Malawi or develop a systematic methodology for assessing barriers in LMICs. For example, in the abstract the authors state: 

"We mixed nine methods using a systems based and a three delays approach (delays in seeking, reaching, or receiving care), to achieve a deep understanding of a trauma care health system and where barrier to access to care are in Karonga, Northern Malawi as an exemplar of a low-income country health system, in addition to methods to use to maximise understanding of trauma systems."

In the introduction the authors state:

"We used a Three Delays framed holistic multi method assessment, to achieve a well-developed understanding of a trauma care health system in a low income country context [….] We also aimed to understand, if taking a frugal approach to data collection which, if any, of the methods would give the most relevant information on the state of the health system to care for injured people."

Simplifying these statements may add clarity. 

* There are also several confusing statements in the methods section. For example:

"Many study methods had their construction or analytical strategy informed by the Three Delays framework[49] with each considered using a framework of barriers to injury care derived from a previously published Delphi study"

Does this mean that the 9 methods selected for this study were developed with the Three Delays Model in mind? This sentence cites reference 49 while the previous citation for the methods was reference 8. Was this Delphi study where the barriers to care examples came from?

* I am not entirely clear about the consensus process. The methods state 

"To assess this data for convergence (the extent to which findings complement or reinforce each other[20]), a consensus exercise was undertaken between three authors [….] Two authors (JW and IE) independently tested the scoring framework against a 40% sample of barriers (each of the Three Delays overall summaries and eight individual barriers) exposed by the study methods."

This could use more explanation. 

* The discussion could be better organized to emphasize key points. Relating to the initial comment regarding the primary purpose of the study, the discussion explores both individual barriers as well as the convergent/divergent nature of different methods. If the primary purpose is to investigate specific barriers to care within Malawi, I would recommend exploring some of the 26 barriers in more depth. If the primary purpose is the development of the assessment than a conversation regarding convergence/divergence is warranted but deserves more details regarding strengths/limitations of different methods as well as their feasibility in implementation. As it stands, both of these are attempted leading to failure to fully explore either. 

For example, if the desire is to review specific barriers, paragraph 9 has a good review of the lack of timely affordable emergency transport and its importance. This could be expanded to provider a more comprehensive overview of this topic.

Similarly, if the goal is to focus on the assessment itself, household surveys, community FGD, and facility process mapping were excellent tools to evaluate Delay 1 barriers. Exploring this finding and why this is would be useful. 

Certain paragraphs could also be removed such as paragraph 8. The authors discuss the importance of financial barriers to care but this does not seem to be supported by their findings and is presented in an unclear manner. Consider removing or phasing in different way.

* The authors do an excellent job incorporating previously literature into their study. While not necessary, additional studies that may be of interest include "Risk Factors for Delayed Presentation Among Patients with Musculoskeletal Injuries in Malawi" or work done by Dr. Sudha Jayaraman in Rwanda. 

Reviewer #3: Alex McConnachie, Statistical Review

I have been asked to provide a review of the use of statistics in the paper by Whitaker et al, an assessment of barriers to access to care after injury in Malawi.

There is very little for me to review. The only statistic reported in the main body of the paper, is a weighted kappa statistic for the grading of strength of evidence, which appears correctly applied and adequately interpreted. Appendix 1 includes assorted descriptive statistics, which seem relevant. Otherwise, there seem to be no other statistical elements to the paper.

What the paper is trying to do seems very worthy, using a range of different methodologies to assess barriers to health care in a particular setting. However, I felt as though none of the individual studies was reported in sufficient detail to fully appreciate any of the findings. Appendix 1 attempts to report key outcomes in support of the grading of evidence, but each report only consists of a sentence or two - not enough to judge whether the assessment is appropriate.

The assessment process itself seemed odd to me. It seems to be conflating "impact" with "quality" of evidence, and labelling this as "strength of evidence of importance". The criteria specified in Table 3 seem clear, though it is hard to say whether they are truly comparable across study methods, and there is no obvious connection with the impact of the barrier (number affected, harm caused). The kappa statistic for independent ratings was quite good, at 0.69, but not particularly high, so I wonder whether having 60% of scenarios assessed by only one assessor is sufficiently robust.

In short, the statistical parts of the paper are fine. However, reading the paper from the perspective of a medical statistician, I felt there was insufficient detail and rigour for the paper to be fully convincing. Saying that, this type of research is not something I would normally be connected with, so I may not be the best judge.

Reviewer #4: Attached.

[LINK]

---

## [Decision Letter · Decision Letter 2]

6 Dec 2023

Dear Dr. Whitaker,

Thank you very much for re-submitting your manuscript "A holistic health system assessment for access to care after injury in low- or middle-income countries: an exemplar study from Northern Malawi." (PMEDICINE-D-22-03524R2) for review by PLOS Medicine.

Thank you for your detailed response to the editors' and reviewers' comments. I have discussed the paper with my colleagues and the academic editor, and it has also been seen again by two of the original reviewers. The changes made to the paper were mostly satisfactory to the reviewers. As such, we intend to accept the paper for publication, pending your attention to the editorial comments below in a further revision. When submitting your revised paper, please include a detailed point-by-point response to the editorial comments.

[LINK]

If you have any questions in the meantime, please contact me (aschaefer@plos.org) or the journal staff on plosmedicine@plos.org.  

We look forward to receiving the revised manuscript by Dec 13 2023 11:59PM.   

Sincerely,

Alexandra Schaefer, PhD

Associate Editor 

PLOS Medicine

plosmedicine.org

Requests from Editors:

ACADEMIC EDITOR COMMENTS

I concur with the reviewer who is asking for recommendations for future research of this type - specifically on which methods produced the most insight and is there a more efficient approach for future studies. Editor’s note: The Editorial team agrees with this comment and feels that the paper would benefit from further discussion of the implications of the study and on recommendations for future research.

FINANCIAL DISCLOSURE

Please revise your Financial Disclosure statement. The funding statement should include: specific grant numbers, initials of authors who received each award, URLs to sponsors’ websites.

TITLE

We suggest changing the title to “Health system assessment for access to care after injury in low- or middle-income countries: A mixed methods study from Northern Malawi.” or similar.

ABSTRACT

1) In the second paragraph of ‘Methods and Findings’ you discuss delay 1, 2 and 3 without previously specifying what delay 1, 2 and 3 are. We suggest adding this information, e.g. in line 73 “For each delay (delay 1 seeking care, delay 2 reaching care, or delay 3 receiving care)..” or similar.

2) l.97: Please change ‘holisic’ to ‘holistic’.

3) The Abstract conclusion section should be more specific to your study results. As written, the conclusion is very general and does not provide an interpretation of the study’s results presented in the abstract, such as the role of the barriers "cost", "transport" and "physical resources”. Please revise to include a more specific and objective interpretation of the data presented in the paper.

AUTHOR SUMMARY

1) Similar to comment #1 under Abstract, in the last bullet point of ‘What Did the Researchers Do and Find?’ it is not clear what delays 1, 2 and 3 signify. Please revise.

2) Please revise the bullet points under the second subheading to ‘What Did the Researchers Do and Find?’, and describe the methods of the study in non-technical language.

3) l.139: Please change ‘holisic’ to ‘holistic’.

4) Please revise the bullet points under the third subheading ‘What Do These Findings Mean?’ as they are currently too similar to the scientific abstract. Also, in the final bullet point of ‘What Do These Findings Mean?’ please describe the main limitations of the study in non-technical language. 

METHODS AND RESULTS

1) ll.214-215: Please provide reference.

2) We suggest adding the definitions of the three delays in the Methods section as they are currently only mentioned in the Introduction.

3) l.343: Please define ‘CI’ at first use.

DISCUSSION

1) ll.461-466: The terms gender and sex are not interchangeable (as discussed in https://www.who.int/health-topics/gender); please use the appropriate term and revise the paragraph.

2) ll.505-506: Please consider revising the wording to improve clarity.

3) ll.544-546: Please provide reference.

4) Please remove the subheading ‘Conclusion’. The conclusion paragraph should be part of the Discussion.

5) In the conclusion paragraph (ll.634-641), similarly to the Abstract, please be more specific to what your study has shown and potential implications. Lines 634-641 are almost a repetition of the Abstract’s conclusions and are rather general statements that can be mentioned earlier on in the discussion.

6) l.637: Please change ‘holisic’ to ‘holistic’.

SUPPLEMENTARY MATERIAL

1) l.139: "Free text responses for other specified answers were reviewed by me." – please specify ‘me’ by indicating author initials.

2) l.187: "we included all to review all VAs in which injury played a role." – please revise.

3) l.260: The terms gender and sex are not interchangeable (as discussed in https://www.who.int/health-topics/gender); please use the appropriate term and revise the entire supplementary material accordingly.

4) p.16/17: Please format the references provided as described in the previous and current revision letter under REFERENCES.

5) Thank you for providing the Good Reporting of A Mixed Methods Study (GRAMMS) checklist. If possible, please format the text to a table and replace the line numbers with paragraph numbers per section (e.g. "Methods, paragraph 1"), since the page numbers of the final published paper may be different from the page numbers in the current manuscript.

6) S4 Table: We noted that some of the data is presented as percentages plus actual numbers, while others are presented only as actual numbers. We suggest revising the table and adding percentages where possible. For example, for "68/228 reported this amongst the top 3 overall barriers. 1st of 6 for delay 2, 3rd of 23 overall." a percentage detail might be helpful. Also, please define "hr/hrs" and make sure all abbreviations are defined.

REFERENCES

1) Please thoroughly revise all references and ensure that journal name abbreviations match those found in the National Center for Biotechnology Information (NCBI) databases (http://www.ncbi.nlm.nih.gov/nlmcatalog/journals), and are appropriately formatted and capitalised (e.g., for reference [2] The Lancet Global Health should be Lancet Glob Health).

2) Please thoroughly revise all references regarding completeness. For example, references [41] or [52] seem to be missing conventional citation details.

3) Where website addresses are cited, please specify the date of access (e.g. "[accessed 2021 Nov 22]").

SOCIAL MEDIA

To help us extend the reach of your research, please provide any X (formerly known as Twitter) handle(s) that would be appropriate to tag, including your own, your coauthors’, your institution, funder, or lab. Please respond to this email with any handles you wish to be included when we tweet this paper.

Comments from Reviewers:

Reviewer #1: I commend the authors for conducting a well designed study on a pertinent topic. This study is the first to address the causes of delay in trauma care specific to Rwanda using rigorous methodology. The are no further changes recommended and it is ready for publication.

Reviewer #2: Thank you for the opportunity to review this revised manuscript. I'd like to commend the authors for their dedication to health system evaluation and conducting an extensive study of the topic in a targeted population. They have made significant changes which have greatly improved the manuscript. I believe the manuscript is suitable for publication but can be enhanced if the authors address the following: 

The authors state that the main purpose of this study is to integrate the findings from multiple methods to develop a holistic understanding of a health system that can be replicated in different settings. As such it would be helpful to provide a grading of the included methods to help future researchers understand which methods to use. For example, GIS evidence did not provide useful information and probably does not need to be included in future work given the limitations outlined in paragraph of the discussion 9. Similarly, do facility assessments and clinical vignettes add significant information that is not found elsewhere? The authors review this in paragraphs 7-11 and have a summary in paragraph 12 but could consider adding their recommendations for which methods to prioritize overall or for each level of delay. A simple table with this information would be useful. I think this would help the authors achieve their main goal which is to create a reproducible method for assessing trauma systems which future studies can build on. 

What are the implications of these results? The study finds that the main barriers are cost, transportation, and physical resources which is well known. What additional information has been provided using multiple methods and how has the use of multiple methods added to evidence-based changes? Given the significant time and monetary investment in conducting a study such as this it is important to provide a reason why others should reproduce this work in similar settings. The study was conducted in 2019-2020 which means there has been amble time to act on these findings. Providing examples of changes that have been made or recommendations of what should be made in the future would strengthen the authors argument that a holistic approach using several different methods was useful. It is difficult to imagine others replicating this work without strong evidence of its superiority compared to less time consuming and cheaper options.

[LINK]

---

## [Decision Letter · Decision Letter 3]

10 Jan 2024

Dear Dr Whitaker, 

On behalf of my colleagues and the Academic Editor, Margaret E. Kruk, I am pleased to inform you that we have agreed to publish your manuscript "Health system assessment for access to care after injury in low- or middle-income countries: A mixed methods study from Northern Malawi" (PMEDICINE-D-22-03524R3) in PLOS Medicine.

I appreciate your thorough responses to the reviewers' and editors' comments throughout the editorial process. We look forward to publishing your manuscript, and editorially there are only a few remaining minor stylistic/presentation points that should be addressed prior to publication. We will carefully check whether the changes have been made. If you have any questions or concerns regarding these final requests, please feel free to contact me at aschaefer@plos.org.

Please see below the minor points that we request you respond to:

1) Table 5: Please define ‘HCW’ and ‘GIS’. Please add the descriptions (Well-aligned, partly well aligned, poorly aligned) to the colored boxes in order to make your figure accessible to those with colour blindness (as done in Table 4).

2) Where website addresses are cited, please add “accessed” before specifying the date of access, e.g. “[accessed 2021 Nov 22]”. Please revise for both the main manuscript and S1 – Unpublished methods supplementary material.

3) S5 Table: Please ensure to introduce and/or define all abbreviations used, such as ‘GIS’, ‘FDG’. Please change “minuntes” to “minutes” in “An administered vignette taking 30 minuntes per participant was fairly time-consuming.”.

PRESS

Sincerely, 

Alexandra Schaefer, PhD 

Associate Editor 

PLOS Medicine